# Generation of Composite *Rosa roxburghii* Plants with Transgenic Roots by *Agrobacterium*-Mediated Transformation

Lisha Gong [1,2], Min Lu [1,2] and Huaming An [1,2,*]

1   Engineering Technology Research Centre for *Rosa roxburghii* of National Forestry and Grassland Administration, Guiyang 550025, China
2   Agricultural College, Guizhou University, Guiyang 550025, China
*   Correspondence: anhuaming@hotmail.com

**Abstract:** *Rosa roxburghii* Tratt. is an emerging fruit endemic to China, which has the reputation of being the "King of Vitamin C" because of its abundance of vitamin C. However, it is also a recalcitrant species that imposes severe limitations on the transformation and whole-plant regeneration processes, restricting the verification of the functional genes. Therefore, developing a feasible and efficient genetic transformation method for *R. roxburghii* is an urgent requirement. Herein, K599 with eGFP was used as the *Agrobacterium* strain to optimize the genetic transformation from four factors: bacterial concentration, seedling age, infection site, and method. First, the original roots of 5-day-old seedlings were excised, and then the slant cuts of the remaining hypocotyls with 0.5 cm length were placed in K599 at an OD600 of 0.4. Subsequently, the explants were planted in a moistened sterile vermiculite after the beveled site was stained with a clump of bacteria. The results showed that the transformation efficiency of this cutting method was almost 28% at 30 days post-inoculation, while the transformation efficiency obtained by injecting 5-day-old seedlings 0.5–1.0 cm away from the primary root with K599 at an OD600 of 0.4 was only about 7%. Taken together, the current findings provide evidence that *Agrobacterium*-mediated transformation is a simple, fast, and efficient approach for generating composite *R. roxburghii* plants. Thus, this method has a broad application to analyze the gene functions in *R. roxburghii* and other related plant species.

**Keywords:** *Agrobacterium*; *Rosa roxburghii* Tratt.; hairy root transformation; K599; composite plants

## 1. Introduction

*Rosa roxburghii* Tratt. is a perennial shrub of the Rosaceae family and an emerging fruit endemic to China; it has become a key fruit tree industry in Guizhou Province [1]. The vitamin C content of the fruit is about 1300–3500 mg·100 g$^{-1}$·FW, which is 455-fold that of apple and 5–22 times that of kiwifruit; thus, it enjoys the reputation of being the "King of Vitamin C" [2,3]. Therefore, the study on *R. roxburghii* has been gradually deepened, especially the whole-genome sequencing by our group (data to be published). In addition, the spatiotemporal expression-specific transcriptome and large-scale cDNA sequencing studies have been conducted [4,5]. However, an efficient transformation method is the prerequisite for functional annotations of the candidate genes. Thus, it is urgent to successfully use appropriate techniques to achieve gene function verification in this plant.

For several years, the methods used to study the functional verification of *R. roxburghii* genes have been limited to model plants or other heterologous expression [2,6]. The transient silencing technology—virus-induced gene silencing (VIGS) on *R. roxburghii*—was not developed until 2021 [7] and was not promoted. In addition, the reprocessing system of *R. roxburghii* has not yet been established, rendering genetic transformation rather challenging. Therefore, the development of a simple, fast, and efficient genetic transformation system of *R. roxburghii* is significant and urgent to interpret and verify the functions of key genes in the biological processes.

In recent years, *Agrobacterium*-mediated hairy root transformation has been on the rise [8], and the resulting hairy roots have been applied to the study of the gene function of various non-model plants [9], such as tea [10], apple [11], pigeon pea [12,13], and cucumber [14]. Moreover, compared with non-tissue culture techniques, the induction of hairy roots by tissue culture is time-consuming and labor-intensive, requiring sterile conditions. Hence, the present study aimed to apply an *Agrobacterium*-mediated transgenic root system to *R. roxburghii* for the first time through non-tissue culture by optimizing from four aspects, bacterial concentration, seedling age, infection site, and infection method, so as to facilitate the functional verification of the genes.

## 2. Materials and Methods

### 2.1. Plant Materials and Agrobacterium Strains

The seeds of *Rosa roxburghii* Tratt. cv. Guinong 5 were obtained from the fruit germplasm resource nursery of the Agricultural College of Guizhou University (Guiyang, China). The clean seeds were stored at 4 °C in the sand, wherein they started to germinate (emerged radicle) after 2–3 months, and then were sown in 7 cm (diameter) × 11 cm (height) pots with sterile soil and were grown in an incubator at 25 °C/22 °C in a 16 h/8 h (day/night) photoperiod under 15,000 Lux with 70% relative humidity.

*Agrobacterium* strain K599 harboring empty vectors, pROK2 and pROK2-eGFP, were a kind gift from Professor Dong Meng from Beijing Forestry University (Beijing, China) and were cultured as described previously [12]. To prepare the inoculation medium, the plates with K599 were incubated at 28 °C for 2 days, and fresh culture K599 was re-suspended using an MES buffer (10 mM MES-KOH, pH 5.2, 10 mM MgCl$_2$, and 100 μM acetosyringone) to adjust the optical density at 600 nm (OD600).

### 2.2. Agrobacterium-Mediated Root Transformation and Generation of Composite R. roxburghii Tratt. Plants

Healthy and uniform seedlings were selected for agro-infiltration using the following injection and cutting procedure based on recent studies [12,14]. In the injecting protocol (Figure 1a), 0.1 mL K599 suspension was injected into the attached plant stem, such that a large drop of infection solution hung on the injection site, followed by a dark day treatment and then covered with a wet sterile vermiculite. After one month of high humidity, the hairy roots developed and the transgenic rate was detected by fluorescence, polymerase chain reaction (PCR), and reverse transcriptase PCR (RT-PCR).

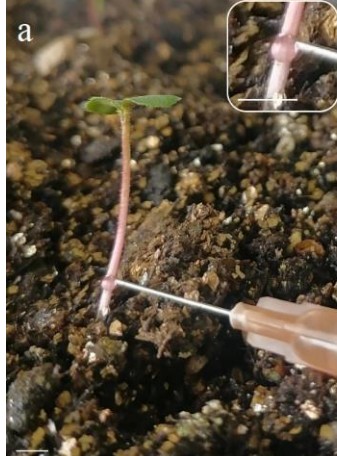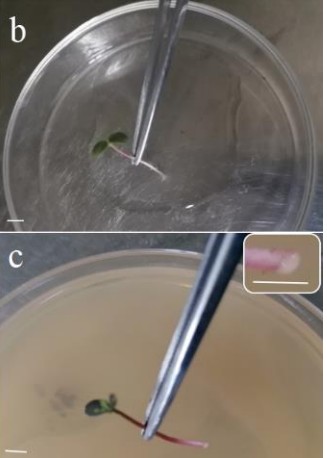

**Figure 1.** *Agrobacterium*-mediated root transformation of *R. roxburghii* achieved by injecting or cutting. (**a**) A healthy 5-day-old seedling by injecting. (**b**) The apical portion of the hypocotyl was cut diagonally in the K599 broth. (**c**) A slant cut of the residual hypocotyl was inoculated on the plate grown on K599 and coated by the bacterial mass. Bars = 0.5 cm.

In the cutting protocol (Figure 1b,c), the primary root was excised from the seedling using a pair of sterile scissors. The apical portion of the hypocotyl was cut diagonally in the K599 broth. A slant cut of the residual hypocotyl was inoculated on the K599 plate and coated with the bacterial mass. The inoculated explant was directly planted into a wet sterile vermiculite and cultured under high humidity. Then, the hairy roots developed adequately at 30 days post-inoculation (dpi).

The efficiency was calculated as follows: Transgenic root induction efficiency = (Positive transgenic hairy root plants/total numbers of injected plants) $\times$ 100%.

### 2.3. Fluorescence Observation

To determine the transgenic and non-transgenic roots, eGFP fluorescence and PCR and RT-PCR analyses were conducted. The eGFP fluorescence (excitation wavelength 460 nm, emission wavelength 500 nm) of *R. roxburghii* plants was detected using a LUYOR-3415RG hand-held lamp (LUYOR, Shanghai, China).

### 2.4. DNA, RNA Extraction and PCR, RT-PCR Analysis

The roots of each transformed seedling were subjected to DNA isolation using the tissue/cell genome DNA isolation kit (spin column) (BioTek, Beijing, China) for PCR analysis. RNA was isolated using the RNAprep Pure plant plus kit (polysaccharides and polyphenolics-rich) (Tiangen, Beijing, China) and reverse-transcribed into cDNA after removing the genomic DNA using PrimeScript™ RT reagent kit with gDNA Eraser (Perfect Real Time) (TaKaRa, Dalian, China) for RT-PCR analysis.

The DNA and RNA were quantified on a NanoDrop spectrophotometer (NanoDrop Technologies, Inc., Wilmington, DE, USA) and analyzed by gel electrophoresis. PCR/RT-PCR amplification was carried out in a 20 μL reaction volume containing 10 μL of 2$\times$ Taq Master Mix*, 0.5 μL of each primer (10 μmol/L), 100 ng of template, and ddH$_2$O. PCR and RT-PCR assays were performed to amplify the eGFP (720 bp). The forward (5′-ATGGTGAGCAAGGGCGAG-3′) and reverse primers (5′-TTACTTGTACAGCTCGTCCA-3′) for the two reactions were the same. The reaction conditions were as follows: 94 °C for 90 s, followed by 30 cycles at 94 °C for 20 s, 58 °C for 20 s, and 72 °C for 60 s, and then 72 °C for 5 min.

### 2.5. Statistical Analysis

Each experiment with over 30 seedlings was a biological replicate and repeated three times. Graphs were plotted using GraphPad Prism dd5 software. Each value represents the mean of three independent experiments with standard devia tion (SD). One -way analysis of variance (ANOVA) and Duncan's multiple range test ($p < 0.05$) were performed using SPSS statistical software (Version 26).

## 3. Results

### 3.1. Establishment of Two Agrobacterium-Mediated Transformation Methods for Generating Composite R. roxburghii Tratt. Plants

In this study, we established two efficient *Agrobacterium*-mediated transformation systems in the *R. roxburghii* Tratt. seedlings. As shown in Figure 2, the constructed vector pROK2-eGFP was introduced into the K599 strains and then infected into 5-day-old seedlings through injection or cutting methods. After about 8 days of growth, a callus appeared around the infection site and expanded gradually. After about 3 weeks, small hairy roots, similar to adventitious roots, grew from the callus (Figure 2a,c). At this stage, we examined whether the target gene was successfully introduced into the hairy roots through fluorescence observation, PCR, and RT-PCR. Compared with the seedlings under natural light (Figure 2a,c), green fluorescence was observed in composite *R. roxburghii* seedlings with transgenic roots under excitation light (Figure 2b,d). The results showed that PCR/RT-PCR-positive roots were consistent with the eGFP-positive types. The PCR results also confirmed that the foreign DNA was inserted into the *R. roxburghii* genome (Figure 2e), while the RT-PCR

analysis demonstrated that the foreign DNA was transcribed in the transgenic hairy roots (Figure 2f). These findings clarified that the two methods provide fast and efficient systems for gene functional analysis, secondary metabolite engineering, and plant stress response studies. As only the roots were transgenic in these systems, it was possible to study the signal transduction between the roots and shoots.

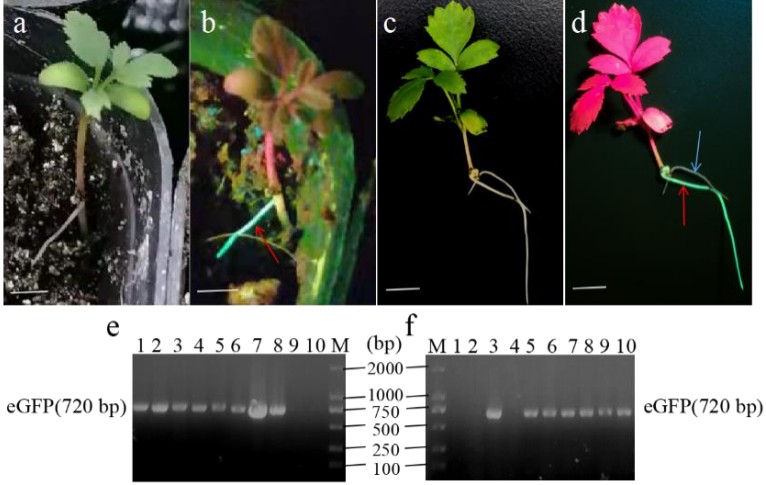

**Figure 2.** Fluorescent observation, PCR, and RT-PCR analysis of transgenic or non-transgenic hairy roots. (**a**,**b**) Composite *R. roxburghii* plant in the natural light or fluorescence irradiation after injection. (**c**,**d**) Composite *R. roxburghii* plant in natural light or fluorescence irradiation after cutting. (**e**) Validation of eGFP integration by PCR. Lines 1–6, eGFP-positive root; line 7, pROK2-eGFP; line 8, K599 (pROK2-eGFP); line 9, eGFP-negative root; line 10, K599 (pROK2). (**f**) Validation of positive roots for eGFP expression by RT-PCR. Line 1, $H_2O$; line 2, K599 (pROK2); line 3, K599 (pROK2-eGFP); line 4, eGFP-negative root; lines 5–10, eGFP-positive root. The red arrow indicated transgenic root; the blue pointed to non-transgenic root. M: 2 kbp Marker. Bars = 0.5 cm.

### 3.2. Optimization of Bacterial Concentration, Seedling Age, Infection Site, and Infection Methods

In order to find the most suitable conditions for *Agrobacterium* K599 to infect *R. roxburghii* seedlings, this study set several factors to optimize the infection system. First, we examined a range of concentrations of *Agrobacterium* solutions (0.2, 0.4, 0.6, and 0.8 OD600 values) for the hairy root transgenic rate. The results showed that the OD600 value of 0.4 was optimal for both the injecting and cutting methods for getting a high hairy root transgenic rates (7% and 28%, respectively) in *R. roxburghii* seedlings, and significant differences were observed between the various bacterial concentrations ($p < 0.05$) (Figure 3).

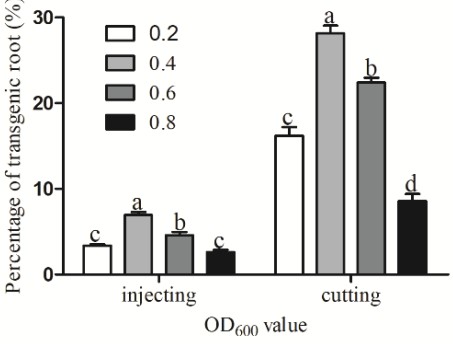

**Figure 3.** Selection of the optimal concentration of the bacterial solution for infection. OD600 values of 0.2, 0.4, 0.6, and 0.8 were experimented with. Effects of different bacterial concentrations on the transgenic root under two infection methods. Letters indicated a significant difference at $p < 0.05$. Values were mean $\pm$ SD for three independent replicates ($n \geq 25$ for each replicate).

Next, we tested the optimal seedling site and age for infection (Figures 4 and 5). The seedlings at three different ages (5-, 10-, 15-, or 20-day-old) were used to determine the optimal age of seedlings for infection (Figure 4a). The results showed that regardless of the method, the transgenic root rate decreased continually with the seedling age (Figure 4b). Especially for the root cutting method, significant differences were noted among the seedling age groups in this experiment. This phenomenon indicated that the younger the seedlings, the better the infection effect. Typically, the optimal seedling age for hairy root induction was 5-days-old. For the injecting method, the stem region close to the hypocotyl was divided into three parts with each being 0.5 cm; these were named positions A, B, and C, respectively (Figure 5a). The transgenic rate of the hairy roots was about 6% for positions B and C, whereas it was significantly lower for A (Figure 5b). On the other hand, the influence of the root cutting position on the transgenic rate was explored, after the original root was excised, according to the factor of the remaining hypocotyl length of the explant as the cutting root site, where 0.5 cm was used as the unit. The results showed that the shorter the remaining hypocotyl length, the higher the infection rate of the transgenic roots (Figure 5c). However, Figure 5d shows that although the transgenic root rate of the seedlings with the remaining 0.5 cm of hypocotyl length was significantly higher than the other two, the seedlings did not grow adequately; in particular, the root lengths were short.

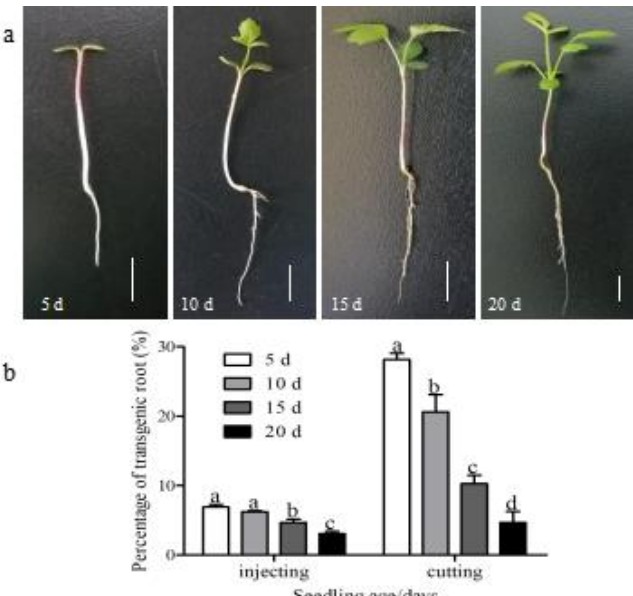

**Figure 4.** Transgenic rate at various seedling ages. (**a**) *R. roxburghii* plants of different seedling ages. (**b**) Effects of seedling ages on the transgenic rates under two infection methods. Letters indicated a significant difference at $p < 0.05$. Values were mean ± SD for three independent replicates ($n \geq 25$ for each replicate). Bars = 1 cm.

Notably, although the two infection methods used in the text have been optimized for their respective systems, the differences in the infection were distinct. Figure 5b,c shows that the infection efficiency of the optimized injection method was lower than that of the root cutting method (7% and 28%, respectively).

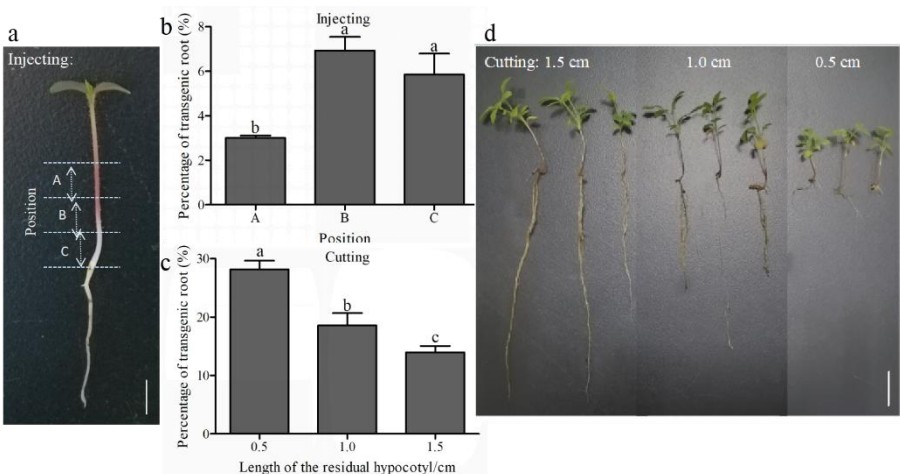

**Figure 5.** Transgenic rate of different infection sites. (**a**) The injection site of 5-day-old seedlings for the injecting method. (**b**,**c**) Effects of the infection site on the transgenic rate in the injecting and cutting methods, respectively. (**d**) Seedling state at 30 dpi after cutting the primary root of 5-day-old seedlings. Letters indicated a significant difference at $p < 0.05$. Values were mean ± SD for three independent replicates ($n \geq 25$ for each replicate). Bars = 0.5 cm.

## 4. Discussion

### 4.1. Establishment of Two Agrobacterium-Mediated Transformation Methods for Generating Composite R. roxburghii Plants

The emerging *Agrobacterium*-induced transgenic technology has been utilized to study the gene function in a variety of plants, providing another possibility for the genetic transformation of plants [10–15]. In recent years, *Agrobacterium*-mediated hairy root culture techniques have been used to develop rapid, simple, and efficient transformation systems for generating stable transgenic roots in vivo, especially two non-tissue culture methods, namely the injecting method [12,13] and the cutting method [14,15]. As a non-model plant, *R. roxburghii* has not yet achieved a breakthrough in its regeneration system and genetic transformation system. Similar to *Camellia sinensis* var. *sinensis* [10], *R. roxburghii* is rich in phenolic substances [16]. Thus, we speculated that the oxidation of phenolic substances released by the explant tissues in *R. roxburghii* and the bactericidal effect of tea polyphenols makes the transformation process difficult and weakens the regeneration ability. In the present study, we induced transgenic hairy roots mediated by the widely applicable *Agrobacterium* K599 in *R. roxburghii* using these two infection methods; the results showed that both methods were feasible (efficiency of 7% and 28%, respectively). The former injecting method has been achieved in nine economic woody plants at an efficiency of 15–39%, including apples of the Rosaceae family [11,12]. However, compared with the other plants, the infection efficiency of K599 by injecting *R. roxburghii* is lower. This phenomenon could be attributed to the tender and slender stems of *R. roxburghii* that cause the stem to break when injected too hard, and the infection is insufficient when injected too lightly. On the other hand, the cutting method has been applied to >80% of vegetable plants [14,15]. However, as *R. roxburghii* in this study is a shrub, it is speculated that non-woody plants are more susceptible to Agrobacterium infection than woody plants, which is in agreement with previous findings [17].

### 4.2. Optimization of Agrobacterium-Mediated Transformation System

The induction rate of *Agrobacterium* is affected by three factors: strains (strain type and concentration), plant (seedling age and infection site), and infection methods. Firstly, the root-causing properties of strains, the donor, are related to the type of Ri plasmid [18,19] and vary markedly [20,21]. The experiments infecting cucumber with K599 and ARqual strains found that the transformation efficiency of the former was about 95%, while that of the latter was 0% [14]. Although K599, namely *Agrobacterium* NCPPB2659, has a weak

pathogenicity [22,23], it has achieved infection in various plants [12]. Hence, this strain was also used to complete the infection of *R. roxburghii* in this study. In addition, the bacterial concentration is also one of the key factors acting on the infection effect [24,25]. Typically, if the concentration is extremely high, a large number of bacterial cells adhere to the surface of the explant, and the bacteria grow rapidly, which inhibits the metabolic process of the plant cells, eventually leading to a decrease in the induction rate or even to the death of the explant; if too low, only a few bacteria can attach to the plant cells to form a sufficient number of hairy roots. A previous study using *Agrobacterium* to infect walnut found that the best effect was when the OD600 was 0.4 [26], which was consistent with the current results.

Secondly, the plant status is critical for receptors. Several studies have compared the induction rates of different seedling ages. For example, Bandaranayake and Yoder [27] discovered that young explants were more successfully induced compared with the old types, and the induction rate decreased significantly with the increasing seedling age. Meng et al. [12] demonstrated that the induction rate of 15- and 30-day-old pigeon pea seedlings were higher than that of the 45-day-old seedlings. These studies have shown that the younger the plant, the higher the induction efficiency, which is similar to our results. The analysis showed that this might be because the cells of these younger plants are in a vigorous period of division and are likely to accept foreign genes, which is conducive to the infection of *Agrobacterium*. Conversely, in most cases, the induction of the efficiency of hypocotyl is higher than the other positions, such as cotyledons. Similarly, Rahman et al. [28] induced the hairy roots of Beta vulgaris by puncture injection and uncovered that the induction efficiency of hypocotyl was higher than that of cotyledons (33% and 8%, respectively). Meng et al. [12] further divided the stem of the hypocotyl into three parts for testing and found that the induction rate was significantly higher in the position closer to the original root of the plant than in the position farther away, which was similar to our results using the injecting method. However, the study on *Carya illinoinensis* had the opposite findings [17]. Thus, we speculated that different parts might have specific effects on the induction rate and genetic transformation of hairy roots due to differences in cell types and endogenous hormone levels.

Thirdly, as bridges, the infection methods are related to the realization of the infection and the simplicity and complexity of the infection process. Herein, two non-tissue culture methods, the injecting and the cutting methods, were used to establish the hairy root transgenic system of *R. roxburghii*, and the transgenic rates were 7% and 28%, respectively, indicating a significant difference, which was in agreement with the previous findings [15,29]. Furthermore, we analyzed the reasons and learned that the cutting method had a greater infection intensity than the injection method. In the cutting method, the slanting cut could increase the surface area for optimum bacterial contact, and the double inoculation with *Agrobacterium* on the cut could increase the possibility of bacterial infection [14,15].

## 5. Conclusions

This study developed a novel, fast, simple, and efficient one-step *Agrobacterium*-mediated transformation system for a non-model plant, *R. roxburghii*, with low contamination risk. Thus, this study provides an alternative to genetic research in the form of various pathways and a scheme to investigate the signaling between the shoot and root of *R. roxburghii*. We speculate that the new method will also improve the transformation efficiencies in other non-model plant species.

**Author Contributions:** Conceptualization, resources, wring—review and editing, H.A. and M.L.; Investigation, methodology, and formal analysis, H.A., L.G. and M.L.; Validation, data curation, wring–original draft preparation, L.G.; Project administration, H.A. All authors have read and agreed to the published version of the manuscript.

**Funding:** This work was supported by grants from the Joint Fund of the National Natural Science Foundation of China and the Karst Science Research Center of Guizhou Province (grant no. U1812401) and the National Natural Science Foundation of China (32260730).

**Acknowledgments:** We are grateful to Dong Meng (Beijing Forestry University, Beijing, China) for kindly providing the *Agrobacterium* K599 strain.

**Conflicts of Interest:** The authors have no relevant financial or non-financial interests to disclose.

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
