# Peer review of "Generation of Composite Rosa roxburghii Plants with Transgenic Roots by Agrobacterium-Mediated Transformation"

_horticulturae, doi:10.3390/horticulturae8111079_

Round 1

Reviewer 1 Report

Please see the highlighted area in the attached document.

 Some minor concerns are:

Italicise the scientific names uniformly throughout the text

Figure legends need to place properly

Reviewer 2 Report

General remarks:

Overall, the manuscript is well written and the methodology seems to be sound, but the novelty of this study is very limited. There are numerous papers reporting Agrobacterium-mediated transformation of Rosa spp. Agrobacterium-mediated transformation as well as particle bombardment are not new for transformation of Rosa spp.. There are several papers describing these methods, including optimizing parameters such as OD of inoculum,… (Debener and Byrne et al. 2014; Shen et al. 2015, Liu et al. 2021). Moreover, the paper repeated the work done on cucumber, in which also the effect of seedling age, cutting method on infection was investigated (Fan et al. 2020, PCTOC)

Major comments:

-        As mentioned above, the novelty is limited. The authors should try to emphasize in what way their paper differs from the other papers. In other words, what is the added value of this paper.

-        The manuscript has too many self-citations (12/37) and papers published in Chinese (10/37), while there are several other relevant papers that were not cited (e.g. Liu et al. 2021).

-        Attention should be given to the naming: A. rhizogenes doesn’t exist anymore, it is either R. rhizogenes or rhizogenic Agrobacterium belonging to one of the genomospecies within the agrobacterium tumefaciens species complex.

-        The authors used a reverse transcriptase PCR (RT-PCR) and not a real-time PCR as mentioned in the paper. Please revise this everywhere

-        A major comment regarding the experimental set-up is that the authors did not include a DNA removal step after RNA extraction and before RT-PCR. The authors cannot rule out that the amplification is from DNA (of residual Agrobacteria that possibly did not infect the plant) or from RNA

-        I’m not convinced of the added value of having separate Fig 1 and 2; I advise to combine them in one figure and only keep the most important panels. Furthermore, the quality (resolution) of most pictures is way too low.

-        Leave out Fig4A. Nothing to see and no added value; also no added value for Fig 5A

-        Some statements are not correct:

o   L263: the authors didn’t analyze the reasons why cutting is more efficient.

o   L269: the authors state that they developed a stable transformation system. However it was not checked if the transformation remained stable in the R. roxburghii plants

Minor comments:

-        Genus and species names should be in Italic

-        Abstract: I would use “transformation efficiency” instead of “transgenic root rate”

-        Layout should be checked (e.g. lines 66-71; 101-104)

-        Figure panel indications (A,B,…) are not correctly positioned

-        Line 202: refs 20-25 instead of 20, 21, 22-25

-        Line 257: infection rate?

-        L276: Patents??

Less...

Reviewer 3 Report

This ms. describes a study on a rose transformation protocol, and compares the importance of various parameters on transformation efficiency. It will be of biotechnology interest in this field.

I do not see any major problem with this ms.

Minor points.

The figure panel labeling (a, b..) is shifted in my version in fig. 1, 2, and 3.

Similarly, in fig. 5, ‘b’ should be positioned further down, and in panel a, ‘10d’, ‘15d’ and ‘20d’ should be positioned correctly.

Line 259 of the text, the sentence “Thirdly…” looks a little bit a tautology to me. Although I understand what is meant, the sentence could be improved.

Round 2

Reviewer 2 Report

I appreciate the changes the authors made. However, the following needs to be addressed before publication:

- K599 is an Agrobacterium strain, but the naming is simply wrong. Look into taxonomy: bv2 strains are renamed to R. rhizogenes, while bv1 strains are Agrobacterium strains. Please adjust this throughout the paper!!

Author Response

Thank you for the positive and constructive comments on our paper entitled “Generation of Composite Rosa roxburghii Plants with Transgenic Roots by Agrobacterium-mediated Transformation”, with manuscript ID: horticulturae-1972782.

Point: - K599 is an Agrobacterium strain, but the naming is simply wrong. Look into taxonomy: bv2 strains are renamed to R. rhizogenes, while bv1 strains are Agrobacterium strains. Please adjust this throughout the paper!!

Response: Thanks for this important comment! And we have adjusted this throughout the paper.